# DDCM: A Computational Strategy for Drug Repositioning Based on Support-Vector Regression Algorithm

**DOI:** 10.3390/ijms25105267

**Published:** 2024-05-12

**Authors:** Manyi Xu, Wan Li, Jiaheng He, Yahui Wang, Junjie Lv, Weiming He, Lina Chen, Hui Zhi

**Affiliations:** 1College of Bioinformatics Science and Technology, Harbin Medical University, Harbin 150000, China; myxu.china@gmail.com (M.X.); liwan@hrbmu.edu.cn (W.L.); j1ah3ng@gmail.com (J.H.); wangyahui@hrbmu.edu.cn (Y.W.); lvjunjie525@126.com (J.L.); 2Institute of Opto-Electronics, Harbin Institute of Technology, Harbin 150000, China; hewm@hit.edu.cn

**Keywords:** drug repositioning, support-vector regression, hybrid matrix, potential therapeutic drugs

## Abstract

Computational drug-repositioning technology is an effective tool for speeding up drug development. As biological data resources continue to grow, it becomes more important to find effective methods to identify potential therapeutic drugs for diseases. The effective use of valuable data has become a more rational and efficient approach to drug repositioning. The disease–drug correlation method (DDCM) proposed in this study is a novel approach that integrates data from multiple sources and different levels to predict potential treatments for diseases, utilizing support-vector regression (SVR). The DDCM approach resulted in potential therapeutic drugs for neoplasms and cardiovascular diseases by constructing a correlation hybrid matrix containing the respective similarities of drugs and diseases, implementing the SVR algorithm to predict the correlation scores, and undergoing a randomized perturbation and stepwise screening pipeline. Some potential therapeutic drugs were predicted by this approach. The potential therapeutic ability of these drugs has been well-validated in terms of the literature, function, drug target, and survival-essential genes. The method’s feasibility was confirmed by comparing the predicted results with the classical method and conducting a co-drug analysis of the sub-branch. Our method challenges the conventional approach to studying disease–drug correlations and presents a fresh perspective for understanding the pathogenesis of diseases.

## 1. Introduction

Drug repositioning is an effective and promising strategy for identifying new therapeutic indications for existing drugs. This approach has the potential to significantly enhance the efficiency and productivity of traditional drug discovery and development, ultimately leading to better health outcomes for patients [1]. Drug repositioning has proven to be a valuable solution for the treatment of certain diseases. While the initial instances of successful drug repositioning were largely serendipitous, there are now several examples of effective repositioning, such as acetylsalicylic acid [2,3], thalidomide [4,5], sildenafil [6,7] and dimethyl fumarate [8,9,10]. As the amount of available biomedical data continues to expand, the benefits of using computational methods to identify potential drug candidates for diseases of interest have become increasingly apparent [11,12]. In the realm of computational drug repositioning, the primary methods include text mining [13,14,15,16,17], network analysis [18,19,20,21,22], and machine learning [23,24,25,26,27,28]. The advantages of computational drug repositioning have been demonstrated by numerous researchers through the development of algorithms and experimental validation.

Text-mining advancements have facilitated the creation of association networks for biomedical entities, like genes, drugs, and diseases [13]. For its capability of integrating multiple data sources, network analysis has been a research hot spot and has been utilized in the field of drug repositioning [18]. Machine learning is ideal for computational drug repositioning, as it can process large amounts of data and help identify associations between biological data [23]. By improving the robustness and accuracy of models, machine learning can lead to more effective drug-repositioning strategies. Machine-learning methods currently rely on drug- and disease-based searching to identify new drug–disease interactions. This strategy has emerged as the main approach for discovering potential drug candidates for diseases [29]. The construction of training and test sets is typically required for machine-learning methods. This involves using known disease–drug relationships to establish benchmarks for evaluating new drug candidates [30]. Both machine-learning and network-analysis methods rely on known disease–drug relationships to construct similarity matrices or networks. These are used to identify and evaluate potential drug candidates [31]. Using existing prior information on known disease–drug correlations when constructing similarity matrices or networks can impact the performance of machine learning and similarity network analysis [30]. In binary classification models, disease–drug relationships that are not present in existing data resources are included in the negative training set. The absence of documented associations does not necessarily mean that the association does not exist in biology. In fact, these yet-to-be-observed associations can be vital pieces of information for discovering new drugs. Furthermore, disease-pathway and drug-mechanism data are essential to understanding the reasons underlying the occurrence and progression of diseases, as well as the mechanisms through which drugs can treat such diseases [32].

As a supervised learning approach, machine learning can typically produce superior results for drug repositioning. DDCM is a machine-learning strategy based on the support-vector regression algorithm. Compared with other methods, DDCM considered more levels of data types, which has certain advantages for model construction. DDCM represented the similarity of related biological characteristics of drugs and diseases in the form of a matrix in the construction process, which ensured biological significance in the training process. The SVR-based DDCM approach covered the advantages of regression models. Unlike traditional classifiers, DDCM could describe the correlation between diseases and drugs in the form of continuous scores. This allowed DDCM to quantitatively characterize the association between disease and drug, overcoming the limitation of the traditional binary classification problem. For the diseases where no therapeutic relationship had been observed, the appropriate correlation score could be given for drug pairs, which solved the problem of negative set matching. This also allowed the DDCM method to maximize the inclusion of drugs that differ significantly from the known therapeutic drugs for the disease within the prediction. The prediction of potential therapeutic drugs for a disease could also be further expanded by the introduction of slack variables in the support-vector regression algorithm. Through a stepwise screening pipeline, the DDCM method could effectively reduce the influence of random error on prediction, and this method has been proven to be stable and flexible in multiple aspects.

## 2. Results

### 2.1. Potential Therapeutic Drugs for 59 Neoplasms

For these diseases in C04.588 branch of Medical Subject Headings (MeSH), the DDCM approach was applied to predict the potential therapeutic drugs (Figure 1A,B).

The distributions of disease–drug correlation scores were relatively mainly distributed around zero; there were slight differences between them. The higher the positive score, the greater the degree of correlation between the drug and the disease, and the more likely the drug was to be a potential therapeutic drug for the disease (Figure 2A). The disease–drug correlation scores changed with the number and combination of diseases involved in the calculation (Appendix A). Therefore, a stepwise screening pipeline was adopted to obtain stable drug candidates and potential therapeutic drugs. The distribution of drug–disease correlation scores was derived for 59 neoplasms under three randomized perturbation types and for 59 neoplasms before randomization. The distribution of disease–drug correlation scores showed different degrees of fluctuation in different types of randomizations (Figure 2B). And, the more diseases were removed from the randomized perturbation experiments, the more fluctuations in the final disease–drug correlation scores were obtained than those when no diseases were removed. The stability score for each drug fluctuated across different random perturbation types. Thus, the intersection of the drugs obtained in each of the three random perturbation types was considered here (Appendix A). Non-small-cell lung cancer (NSCLC) was an example. The stable drug candidates for non-small-cell lung cancer were screened by a stepwise screening pipeline. The stability scores of the 50 drugs to be screened in different states of random perturbation are shown (Figure 2C). The closer the stability score is to one, the more stable the drug is under such conditions. The stability scores in several random perturbation conditions were close to one, indicating that the volatility of the drug’s stability was low and more likely to be a potential therapeutic drug. The stepwise screening pipeline screened drugs with high stability and a low degree of stability fluctuation in different random states, resulting in 25 potential therapeutic drugs for NSCLC. There were potential therapeutic drugs for neoplasms other than gallbladder cancer and inflammatory breast carcinoma. These diseases were each predicted to range from 11–30 potential therapeutic drugs (Figure 2D). (Appendix A).

### 2.2. Potential Therapeutic Capacity Analysis and Validation of Potential Therapeutic Drugs

The effectiveness of potential therapeutic drugs was evaluated in the literature aspect, the functional aspect, the drug-target aspect, and the survival analysis aspect.

#### 2.2.1. The Literature Aspect

From the literature aspect, text-mining techniques were leveraged by mining keywords, whether a potential therapeutic drug for a disease has been investigated for the treatment of the disease, whether a potential therapeutic drug acts in combination with other drugs for the treatment of a disease, or whether the drug target of a drug has been shown to be a potential therapeutic target for the disease. For NSCLC, the literature validation rate of the screened potential therapeutic drugs reached 56.0% (14/25) (Appendix A).

#### 2.2.2. The Functional Aspect

From the functional aspect, since the functional properties of drugs were such that they exert their effects through target genes, we focused on the effects of drugs through their target genes. In turn, known disease pathogenic genes can reveal pathogenesis. Thus, functional enrichment was performed for drug-target genes of potential therapeutic drugs and drug-target genes of known drugs for each disease.

We focused on genes associated with non-small-cell lung cancer, including drug targets for potential therapeutic drugs, drug targets for known drugs, and pathogenic genes. In an attempt to reveal the therapeutic mechanism of potential therapeutic drugs, these genes were mainly examined. This part of the gene was categorized into four groups (Group I consisted of five genes: genes in the overlapping part of drug targets of potential therapeutic drugs and drug targets of known drugs and pathogenic genes. Group II consisted of seven genes: genes in the overlapping part of drug targets of potential therapeutic drugs and drug targets of known drugs. Group III consisted of ten genes: genes in the overlapping part of drug targets of potential therapeutic drugs and pathogenic genes. Group IV consisted of thirty-six genes: drug targets of potential therapeutic drugs only) (Figure 3A). An enrichment analysis was performed separately for potential therapeutic drugs and drug targets of known therapeutic drugs for NSCLC (Figure 3B,C).

Functional enrichment was performed for each set of genes. Partial results of some major genes in each group are shown (Figure 3D–G). (Appendix A).

In Group I, where the mechanism of pathogenesis and the potential therapeutic mechanism were similar to the known therapeutic mechanism, the literature validation rate of potential therapeutic drugs was 50.0% (3/6). In Group II, where the mechanism of treatment was similar to that of known therapeutic drugs, the literature validation rate of potential therapeutic drugs was 66.7% (4/6). In Group III, where the mechanism of pathogenesis and potential therapeutic mechanism were similar, the literature validation rate of potential therapeutic drugs was 75.0% (3/4). In Group IV, where the potential therapeutic mechanism was specific, the literature validation rate of potential therapeutic drugs was 70.0% (7/10).

The results of the enrichment analysis showed that, for the drug targets of potential therapeutic drugs for NSCLC, all groups of genes were enriched in peptidyl–tyrosine phosphorylation (GO:0018108), peptidyl–tyrosine modification (GO:0018212), positive regulation of kinase activity (GO:0033674), and other such GO terms. The enrichment of drug-target genes for known drugs in NSCLC can be seen in the enrichment of some genes to these GO terms as well. It was known that Groups III and IV do not overlap with the drug-target genes of known drugs. Genes in Group IV do not overlap with the pathogenic genes, which were similarly enriched in a common functional class with the drug targets of known drugs, suggesting that these potential therapeutic drugs, although targeting different genes than known drugs, may perform the same or similar functions through different biological pathways.

Klarisa Rikova et al. characterize tyrosine kinase signaling across 41 non-small-cell lung cancer (NSCLC) cell lines and over 150 NSCLC tumors. Profiles of phosphotyrosine signaling are generated and analyzed to identify known oncogenic kinases, such as EGFR and c-Met, as well as novel ALK and ROS fusion proteins. By focusing on activated cell circuitry, the approach provides insight into cancer biology not available at the chromosomal and transcriptional levels and can be applied broadly across all human cancers [33]. The lymph node metastasis of NSCLC exclusive somatic mutation (LME-SMs) genes was enriched in peptidyl–tyrosine phosphorylation, peptidyl–tyrosine modification, protein tyrosine kinase activity, and transmembrane receptor protein kinase [34]. Santos GdC et al. have proven that tyrosine kinase inhibitors (TKIs) are especially effective in patients whose tumors harbor activating mutations in the tyrosine kinase domain of the *EGFR* gene. More recent trials have suggested that, for advanced NSCLC patients with *EGFR* mutant tumors, initial therapy with a TKI instead of chemotherapy may be the best choice of treatment [35]. And since the drug-target genes of these groups were enriched with the drug-target genes of known drugs on these GO terms, it can be inferred that the potential therapeutic drugs can be therapeutic drugs for NSCLC from the functional aspect. As a whole, the drug targets predicted for potential therapeutic drugs for NSCLC can play a corresponding role in the functional aspect. In terms of each drug individually, the corresponding drugs were also useful for the treatment of NSCLC from different perspectives.

In the results of drug-target enrichment analysis for Groups II, III, and IV, we can find that these drug targets were enriched to the function of protein autophosphorylation (GO:0046777) in addition to the common three functional classes, indicating that potential therapeutic drugs and known therapeutic drugs also play a role in the treatment of disease through the pathogenic mechanism of the disease. Jing-Qiang Huang et al. suggested that the SRPK1/GSK3β axis promotes gefitinib resistance by activating the Wnt pathway in the form of autophosphorylation and may serve as a potential therapeutic target for overcoming gefitinib resistance in NSCLC [36].

For the additional enrichment of Groups II and IV to the common function positive regulation of collagen metabolic process (GO:0010714), potential therapeutic drugs and known therapeutic drugs have a similar therapeutic pattern. It is possible that the therapeutic effect was exerted by affecting this biological pathway. Hao Chang et al. suggested that the overexpression of collagen XVIII was associated with NSCLC progression and a poor outcome. Thus, collagen XVIII expression may serve as a useful prognostic marker in patients with NSCLC [37].

For Groups III and IV jointly enriched beyond the three GO terms, we identified positive regulation of phosphatidylinositol 3-kinase activity (GO:0043552), potential therapeutic drugs that exert a therapeutic effect on the pathogenic mechanism of the disease, and this therapeutic effect is unique relative to known therapeutic drugs. Panpan Zhang et al. proved when phosphatidylinositol 3-kinase/protein kinase B signaling was inhibited by corresponding inhibitors, PD-L1 expression was downregulated, and apoptosis was upregulated in the cisplatin-treated cancer cells. These results suggest that the upregulation of PD-L1 promotes a resistance response in lung cancer cells that might be through activation of the phosphatidylinositol 3-kinase/protein kinase B pathway and suppression of tumor-infiltrating lymphocytes [38].

In addition, for Group IV genes (*BTK*/*LYN*), the enriched GO term toll-like receptor signaling pathway (GO:0002224) was shown to be associated with NSCLC. Toll-like receptors (TLR) recognize pathogen molecules and danger-associated signals that stimulate inflammatory processes. TLRs have been studied mainly in antigen-presenting cells, where they exert important immune regulatory functions, but they were also expressed by epithelial tumor cells, where they have been implicated in tumor progression. Saradiya Chatterjee et al. proved that, in patients with non-small-cell lung cancer, expression analyses revealed that high TLR7 expression was strongly associated with resistance to neoadjuvant chemotherapy and poor clinical outcomes [39]. *BTK* was targeted by DB01254, and *LYN* was targeted by DB01254 and DB08901, suggesting that both drugs may exert the pharmacological effects of drugs from the perspective of inhibiting the high expression of toll-like receptors.

#### 2.2.3. The Drug-Target Aspect

For the known drug-target genes and drug targets of potential therapeutic drugs for non-small-cell lung cancer, according to KEGG Mapper, most of their drug targets are involved in cancer-related pathways, both for known therapeutic drugs and potential therapeutic drugs. And, the drug targets of potential therapeutic drugs (including four groups of genes) are mainly involved in the PI3K-Akt signaling pathway and the Jak-STAT signaling pathway of the cancer pathway (Figure 4).

Except for Group I, which was available for all aspects of genes, when a drug targets genes in Groups II, III, and IV, the drug should have a similar therapeutic mechanism to known drugs and be associated with the pathogenic mechanism of the disease. And in some aspects, it has its own unique therapeutic mechanism. For example, ponatinib (DB08901) is a kinase inhibitor used to treat patients with various types of chronic myeloid leukemia (Drugbank: ponatinib). These signaling pathways play an important role in the development of cancer. Ren M et al. demonstrated that pharmacological inhibition of FGFR1 kinase activity with ponatinib may be effective for the treatment of lung cancer patients whose tumors overexpress *FGFR1* [40]. Ponatinib targets *FGFR2* and exerts the same therapeutic effect as known drugs for NSCLC. *FGFR2* and other FGFR kinase family gene alterations have been found in both lung squamous cell carcinoma and lung adenocarcinoma [41]. Also, ponatinib targets genes in parts that overlap with pathogenic genes such as *KDR*. The clinical outcomes of patients with advanced NSCLC receiving first-line bevacizumab plus chemotherapy regimens might be impacted by polymorphism V297 L through mediating the mRNA expression of *KDR* [42]. In addition, ponatinib targets potential therapeutic drug-specific targets such as *ABL1*. Mutations in *ABL1* identified in primary NSCLC tumors and a lung cancer cell line increase downstream pathway activation compared to wild-type *ABL1* [43].

When a drug targets genes in Groups III and IV, the drug should have its own unique therapeutic mechanism for the pathogenic mechanism of the disease. For example, dasatinib (DB01254) is an oral dual BCR/ABL and SRC family tyrosine kinase inhibitor approved for use in patients with chronic myelogenous leukemia (DrugBank: dasatinib). It inhibited migration and invasion, induced cell-cycle arrest, and partial apoptosis in NSCLC cell lines. Dasatinib is a drug with multiple targets (these target genes are mainly located in Groups III and IV), which works by inhibiting the metastasis of epithelial cells and affecting related signaling pathways to treat non-small-cell lung cancer. It was demonstrated that this drug will be a unique therapeutic idea for potential therapeutic drugs in NSCLC. Molecular alterations of *YES1*, a member of the SRC, can be found in a significant subset of patients with lung cancer. Moreover, high YES1 protein expression was an independent predictor for poor prognosis in patients with lung cancer [44]. Phase I/II studies have shown that dasatinib in combination with erlotinib is safe and feasible for the treatment of NSCLC [45,46]. Some other targets targeted by dasatinib, such as *PDGFRB* [47] and *MAPK*, have been validated in the literature accordingly.

Romidepsin (DB06176) is a histone deacetylase (HDAC) inhibitor used to treat cutaneous T-cell lymphoma (DrugBank: romidepsin). The drug targets of the drug were mainly in Group IV and Group II. Theoretically, the therapeutic mechanism of the drug has similarities to known drugs as well as its own unique aspects. The drug has some similarities with the known drugs for NSCLC in terms of the mechanism of cure. There were also some individual therapeutic mechanisms (e.g., branching morphogenesis of an epithelial tube, cell–substrate adhesion, etc.). Karthik S et al. proved that romidepsin and bortezomib cooperatively inhibit A549 NSCLC cell proliferation by altering the histone acetylation status and the expression of cell-cycle regulators and MMPs. Romidepsin along with bortezomib might be an effective treatment approach for A549 NSCLC cells [48]. This drug also has an impact on improving the chemosensitivity of NSCLC by targeting the *ABCC1* gene [49,50].

In addition to this, if a drug targets only Group IV, the potential therapeutic drug may have an impact on the treatment of the disease from a unique therapeutic mechanism. Alisertib (DB05220) is a novel aurora A kinase inhibitor under investigation for the treatment of various forms of cancer (DrugBank: alisertib). Alisertib is an oral aurora kinase inhibitor that has been shown to induce cell-cycle arrest and apoptosis in preclinical studies. It is currently under investigation for a wide variety of malignancies, including hematologic and solid tumors [51]. Wang et al. found that an AURKA inhibitor, alisertib, treatment restored the susceptibility of resistant cells to EGFR-TKIs and partially reversed the EMT process, thereby reducing migration and invasion in EGFR-TKI-resistant cells. This study provides evidence that targeting the AURKA signaling pathway by alisertib may be a novel approach for overcoming EGFR-TKI resistance and for the treatment of metastatic EGFR-TKIs in NSCLC patients [52].

Among the potential therapeutic drugs for NSCLC, there are some drugs that are temporarily unproven in the literature.

Flucytosine (DB01099) is an antifungal indicated only to treat severe infections throughout the body caused by susceptible strains of Candida or Cryptococcus (DrugBank: flucytosine). Flucytosine targets the *DNMT1* gene in Group I. *DNMT1* is mainly enriched to the GO:0033002 functional class, namely muscle-cell proliferation, and Jia et al. demonstrated that NSCLC-derived exosomes promote cell proliferation and inhibit cell apoptosis in both normal lung fibroblasts and NSCLC cells by delivering ASMA. High expression of DNMT1 protein in serum may increase the pathogenesis of non-small-cell lung cancer and may play an important role in the early development of lung cancer. Genes in Group I were mainly enriched to some cancer-related signaling pathways, immune responses, and cell-differentiation-related functions. Some of the literature has confirmed the association of these functional classes with non-small-cell lung cancer [53].

Terazosin (DB01162) is an alpha-1 adrenergic antagonist used in the treatment of symptomatic benign prostatic hyperplasia and the management of hypertension (DrugBank: terazosin). Terazosin targets *TGFB1* in Group III and *ADRA1A*, *ADRA1B*, *ADRA1D*, *KCNH2*, *KCNH6*, and *KCNH7* in Group IV. For *TGFB1*, Ki-Eun Hwang confirmed that TGF-β1 induces EMT, which leads to lung cancer cell migration and invasion. It is possible to treat non-small-cell lung cancer by inhibiting this aspect of the *TGFB1* gene [54].

Colchicine (DB01394) is an alkaloid used in the symptomatic relief of pain in attacks of gout and to treat the inflammatory symptoms of Familial Mediterranean Fever (DrugBank: colchicine). Colchicine targets the *TUBB* gene in Group I. Knockdown of *TUBB* sensitizes cells to MTAs, while overexpression confers resistance. A high expression of *TUBB* is correlated with worse survival of lung adenocarcinoma [55]. The *TUBB* gene is primarily enriched for immune-related neutrophil activity. Proinflammatory cytokines are centrally involved in tumor progression and survival in non-small-cell lung cancer, and both the presence of infiltrating neutrophils and bacterial infection in the lung may indicate a poor prognosis. Interaction between bacterial pathogens, neutrophils, and tumor cells might amplify the release of proinflammatory cytokines, which may promote tumor growth in vivo [56].

#### 2.2.4. The Survival-Essential Gene Aspect

Furthermore, survival analysis is widely used in clinical and epidemiological research. Its use in the contemporary medical literature is widespread [57]. The Cancer Dependency Map (DepMap; https://depmap.org/portal/; 21Q4 Public+Score, Chronos, released on 3 November 2021) utilizes RNA interference (RNAi) and Clustered Regularly Interspaced Short Palindromic Repeats (CRISPR-Cas9) technologies to screen for the necessity of various genes for tumors. DepMap analyzes hundreds of cancer cell-line models to obtain information on the genome of individual cell lines and their sensitivity to genetic and small molecule perturbations. Multiple analyses and studies of tumor cells were attempted to identify the genetic and pharmacological dependence of tumors as well as predict their biomarkers. Based on the potential therapeutic drugs, we performed the identification of survival-essential genes for each group of genes for these neoplasms. For NSCLC, several subgroups of drug targets for potential therapeutic drugs can be achieved by a CRISPR knockout screen, which represents the impact of knocked-out genes on the survival of cancer cell lines [58]. Typically, the cut-off value will be set to −0.5 for gene-effect scores, indicating a significant depletion of cell lines [59].

The gene effects of 94 NSCLC cell lines derived from the genome-wide CRISPR knockout screens were stored in the DepMap database. Each group of genes was screened for survival-essential genes by this method. A total of 58 genes were involved in the identification, of which 17 were essential for survival (Figure 5) (Appendix A). Targeting these genes, which were essential for survival, could have the effect of treating neoplasms Further, we screened for drugs corresponding to these survival-essential genes (Table 1). We determined that these drugs that target survival-essential genes were more likely to have an impact on the survival of cancer cell lines and were more likely to be therapeutic drugs for NSCLC.

Among the potential therapeutic drugs corresponding to these survival-essential genes, the literature validation rate was 70.0% (7/10). Among them, the literature validation rate of potential therapeutic drugs in Group I was 50.0% (2/4), in Group II was 50.0% (2/4), in Group III was 100.0% (3/3), and in Group IV was 100.0% (4/4).

Podofilox (DB01179) is a topical agent used for the treatment of external genital warts and perianal warts. *TOP2A* and *TUBB* targeted by podofilox were located in Groups I and II, respectively, and these two genes were identified as survival-essential genes in 94 cell lines and 88 cell lines, respectively. The description of podofilox in DrugBank is that it may have antineoplastic properties, as do some of its congeners and derivatives. (DrugBank: podofilox). Guo et al. proved that podophyllotoxins (podofilox), including epipodophyllotoxin derivatives, can act on a diverse array of drug targets in cancer cells and, thus, possess potent activity against various forms of cancer cell lines, including drug-resistant forms [60]. Likewise, flucytosine (DB01099) targets *DNMT1* in Group I, which was considered to be a survival-essential gene in 80 cell lines. It has been combined with newer azole antifungal agents. It also plays an important role in a new approach to the treatment of cancer [61]. Colchicine (DB01394) targets *TUBB* in Group II, which was considered a survival-essential gene in 88 cell lines. It was concluded that an untapped potential exists for exploiting the colchicine scaffold as a pharmacophore, with the possibility of increasing its affinity for tubulin isotypes overexpressed in cancer and decreasing it for normal cells, thereby widening the therapeutic window [62].

Based on the performance of the drug targets of these drugs in knockout screens for survival-essential genes, as well as descriptions in DrugBank and various studies suggesting that although these drugs have not been directly demonstrated in the literature to have therapeutic effects in NSCLC, their pharmacological effects make them promising potential therapeutic drugs for NSCLC.

From a survival perspective, we can conclude that the genes targeted by certain unproven drugs (DB01099, DB01179, DB01394, etc.) in the literature may be survival-related genes. Thus, these unproven drugs prove their importance not only from a functional point of view but also from the point of view of targeting genes essential for survival, suggesting that such drugs may be potential therapeutic drugs.

### 2.3. Drug Research of Similar Neoplasms in MeSH Branches

Usually, similar diseases tend to have the same susceptible genes or therapeutic targets [63,64] and similar drugs combined with similar drug targets [65]. MeSH provides a consistent way to find content with different terms but the same concepts. MeSH organizes its descriptors into a hierarchy. Diseases under the same branch have a higher similarity. To further confirm the validity of the DDCM approach, one of the sub-branches of neoplasms was chosen. These four diseases were central nervous system cancer, brain cancer, supratentorial cancer, and pituitary cancer in the same branch based on the stepwise relationship of the MeSH branch (Figure 6A). All four diseases have some overlap in terms of known therapeutic drugs and potential therapeutic drugs. Some of these potential therapeutic drugs were even confirmed in certain aspects. The co-drugs analysis of the MeSH sub-branch has also demonstrated the validity of the DDCM approach from the other side.

Considering the genetic aspect, the overlap among the drug-target genes of the screened potential therapeutic drugs, the target genes of the known drugs, and the pathogenic genes of the diseases was examined for the four diseases. It could be found that the degree of overlap between the screened potential therapeutic drugs and the drug targets of the known drugs for these four similar diseases was relatively large, while the degree of overlap with the pathogenic genes was smaller (Figure 6B(a–d)). This indicated that the therapeutic mechanisms of both screened potential therapeutic drugs and known therapeutic drugs were similar, while they were different from the pathogenic mechanisms of the diseases. In addition, the drug targets of the known therapeutic drugs, the drug targets of the screened potential therapeutic drugs, and the pathogenic genes of the four diseases overlap to some extent (Figure 6C–E), indicating that the four diseases were similar in their respective therapeutic and pathogenic mechanisms. This was further illustrated by the similarity of the four diseases, since they were taken from the same branch of MeSH. Combined with the previous results, similar diseases have drugs in common, so potential therapeutic drugs for these diseases and known therapeutic drugs should each have some degree of overlap.

The analysis of known therapeutic drugs and potential therapeutic drugs for these diseases showed that there was indeed some degree of overlap between the known therapeutic drugs for these diseases. Also, by predicting potential therapeutic drugs for these diseases, common therapeutic drugs could be found between these four cancers (Figure 6F,G).

Among the screened potential therapeutic drugs for the four diseases, plicamycin (DB06810) was one of the screened potential therapeutic drugs common to central nervous system cancer, brain cancer, supratentorial cancer, and pituitary cancer. Plicamycin is an antineoplastic antibiotic produced by Streptomyces plicatus (DrugBank: Plicamycin); 6-O-benzylguanine (DB11919) was predicted to be the potential therapeutic drug for brain cancer, central nervous system cancer, and supratentorial cancer [66]. Many current experimental studies have demonstrated the usefulness of the drug in the treatment of brain cancer, central nervous system cancer, and other conditions [67,68,69,70]. The co-drug analysis of potential therapeutic drugs for diseases of the same MeSH branch not only strengthened the conclusion of co-drugs for similar diseases but also highlighted the validity of the method laterally.

It was observed that four genes, *AHR*, *ABCC1*, *ABCC2*, and *NR1I2*, were the targets of the screened potential therapeutic drugs for all four diseases with a higher degree of overlap compared to the known drug targets. From these four drug-target genes, potential therapeutic drugs were traced from different disease perspectives (Figure 6H). Among the four genes, we mainly focused on the *AHR* gene. Emerging evidence suggests the promoting role of the *AHR* in the initiation, promotion, progression, invasion, and metastasis of cancer cells [71]. *AHR* activates the expression of multiple phase I and II xenobiotic chemical metabolizing enzyme genes (such as the *CYP1A1* gene) involved in cell-cycle regulation. Beta-Naphthoflavone (DB06732), which targets *AHR*, has been shown to regulate cell differentiation in the central nervous system [72].

## 3. Discussion

The computational drug-repositioning approach is a successful and promising technique to efficiently discover potential novel therapeutic drugs for diseases and new indications for existing drugs. The potential therapeutic drugs obtained can expand the therapeutic drugs available for the disease and expand the more multifaceted indications of known drugs. We constructed an SVR-based DDCM strategy to predict potential therapeutic drugs for the diseases. The approach was successfully applied in the prediction of potential therapeutic drugs for 59 diseases in the MeSH neoplastic branch. The number of potential therapeutic drugs ranges from 11 to 30. Some well-proven drugs emerged by analyzing the feasibility of potential therapeutic drugs in terms of the literature, function, drug targets, and survival-essential genes, respectively. And we also found similar drugs for diseases in the subbranches of the MESH branch.

In the strategy, SVR calculated continuous correlation scores between diseases and drugs through learning and fitting. The distribution of disease–drug correlation scores predicted by the SVR algorithm was roughly distributed on both sides of zero, but the overall distribution was different from each other. It was believed that in the process of constructing the hybrid matrix, the similarity between diseases and diseases, and the similarity between drugs and drugs was calculated by the design vector, and the number of disease pathogenic genes for each disease or drug-target genes for each drug was different. So, the calculated correlation scores were somewhat different. Thus, we considered the disease–drug correlation scores for each disease separately and conducted subsequent screening operations to derive potential therapeutic drugs instead of performing a cross-sectional comparison to take a percentage threshold as a screening condition for drug candidates. For each disease, after a step of stratified screening, the relevant drug obtained has a higher and more stable correlation with the disease, and we believe that drugs at this point already have the possibility to be used as potential therapeutic drugs for the disease.

Compared with traditional machine-learning methods, such as support-vector machine (SVM), K-nearest neighbor (KNN), and random walk (RW), DDCM methods have certain advantages in terms of performance (AUC = $9.96\times 10^{-1}$) (Appendix A). Similar to the results of Gottlieb et al. [27] in predicting potential therapeutic drugs for colon cancer, the PREDICT method of Gottlieb et al. and the DDCM approach simultaneously predicted tioguanine (DB00352), fludarabine (DB01073), dasatinib (DB01254), colchicine (DB01394). As well as predicting the potential therapeutic drugs for testicular cancer and comparing the PREDICT method, we found that the commonly predicted drugs were colchicine (DB01394), idarubicin (DB01177), etc. This also proved that the predicted results of our method were convincing to some extent; these drugs might have the potential to treat the corresponding diseases to some extent.

We confirmed the generalizability of the DDCM approach by predicting potential therapeutic drugs for diseases in other branches of the MeSH. In the prediction of potential therapeutic drugs for 49 vascular diseases, the number of drugs predicted ranged from 14 to 26. We likewise found drugs that were well-validated (Appendix A). An analysis of genes associated with cerebrovascular disease revealed that the genes associated with cerebrovascular disease were divided into several gene families, and these drug-targeted gene families have been demonstrated in the literature to be associated with the treatment of cerebrovascular disease. A typical example was tranylcypromine, which can modulate monoaminergic transmission. It suggests that this drug is a promising lead compound for the further development of drugs to be used in therapy for cerebrovascular and neurological diseases. Similarly, in vascular disease, we analyzed potential therapeutic drugs for a particular branch of the disease and found common potential and known therapeutic drugs between the branches. The prediction of vascular diseases further validated the feasibility and generalizability of the DDCM.

## 4. Material and Methods

To screen potential drug candidates for the treatment of specific diseases, we developed the DDCM approach. A hybrid matrix of disease–drug correlative relationships was constructed by assessing the similarity of multiple levels for diseases and drugs. For a specific disease, the support-vector regression algorithm was used to calculate the disease–drug correlation score and to further screen the potential therapeutic drugs with stable predictions for that disease by setting stability criteria and potential therapeutic criteria. The effectiveness of potential therapeutic drugs was assessed at the levels of the literature, function, drug target, and survival-essential genes.

### 4.1. Data Retrieval

Drug-target gene data from the DrugBank database (https://go.drugbank.com/; version 5.1.8, released on 3 January 2021) [66] were selected. Meanwhile, drug-pathway data were obtained from the Comparative Toxicogenomics Database (CTD, http://ctdbase.org/; released on 5 August 2021) [73] and integrated with drug-related pathways from the Kyoto Encyclopedia of Genes and Genomes (KEGG, https://www.genome.jp/kegg/; version 98.1, released on 1 May 2021) [74]. The drug names in this study were unified with the drug IDs from the downloaded DrugBank to obtain a list of drug names.

Disease–drug correlation data were obtained from the CTD and DrugBank databases, and disease–drug relationship pairs predicted by bioinformatics methods were removed. Next, pathogenetic genes were obtained based on the integration of Online Mendelian Inheritance in Man (OMIM, https://omim.org/; released on 28 May 2021) [75], Disease Ontology (DO, https://disease-ontology.org/; released on 11 September 2018) [76], and CTD databases from previous studies, where data described as pathogenic genes in the above databases were integrated into the collection of pathogenic genes we used. On the other hand, integrations were obtained for the pathways associated with these diseases in the CTD and KEGG databases. The disease names were aligned with the disease DOIDs in the DO database.

The data types and data sizes in the dataset are given in Appendix A.

### 4.2. Strategies for Evaluating the Degree of Disease–Drug Correlation

An important step in the DDCM approach is the computation of drug–disease correlations, which was accomplished by separately computing the similarity between each drug and disease.

#### 4.2.1. Construction of Hybrid Matrix for Input SVR

Here, a hybrid matrix containing both disease and drug similarity information was obtained by using the adjusted cosine similarity (ACS) method to calculate drug similarity and disease similarity. The hybrid matrix by the operation of matrix splicing with the obtained drug similarity and disease-similarity information was gained. The support-vector regression algorithm (SVR algorithm) was then used to measure the degree of correlation between a disease and drugs.

1.Construction of Disease-Similarity Matrix and Drug Similarity Matrix

To construct the hybrid matrix for input to the SVR algorithm, the drug similarity and disease similarity were calculated separately as the basis. The correlations between diseases and drugs were calculated separately using the ACS.

For each disease, a feature list could be constructed whose length was the number of all genes. Each position in the list corresponds to a gene, and the position corresponding to the pathogenic gene for the disease was marked as 1, and the rest of the positions were marked as 0. The ACS of the gene signature lists corresponding to the two diseases was regarded as the similarity of the pathogenic genes of the two diseases. Similarly, the similarity of the disease-pathway data was calculated using the ACS. The mean value of similarity at both the pathway and gene levels was used as the final similarity for two diseases (Appendix A).
(1)$$\begin{array}{c} Di_{g}=\left\{Di_{g1},Di_{g2},Di_{g3},\ldots ,Di_{ga},\ldots ,Di_{gk}\right\},\forall a\in (1,k),\forall Di_{ga}\in \left\{0,1\right\}\\ Di_{p}=\left\{Di_{p1},Di_{p2},Di_{p3},\ldots ,Di_{pb},\ldots ,Di_{ph}\right\},\forall b\in (1,h),\forall Di_{pb}\in \left\{0,1\right\} \end{array}$$
where $Di_{g}$ and $Di_{p}$ represent the vectors of disease genes and disease pathways, respectively, and k and h represent the lengths of the two vectors, respectively.

To avert the similarity measures obtained based on different distance scale functions may produce different data distribution patterns, the next step was considered to be to jointly construct a disease similarity and drug similarity based on two levels based on disease similarity and drug similarity. To ensure that the data from the similarity of the two levels in the hybrid matrix can be compared together, the ACS was again chosen to measure the similarity of the drugs. Similarly, the calculation method of the disease-similarity matrix in the previous section was chosen to calculate the similarity of the two levels of data of the drug-target genes and drug-related pathways, respectively. The mean value of similarity at both the pathway and gene levels was used as the final similarity for two drugs (Appendix A).
(2)$$\begin{array}{c} Dr_{g}=\left\{Dr_{g1},Dr_{g2},Dr_{g3},\ldots ,Dr_{gc},\ldots ,Dr_{gc}\right\},\forall c\in (1,r),\forall Dr_{gc}\in \left\{0,1\right\}\\ Dr_{p}=\left\{Dr_{p1},Dr_{p2},Dr_{p3},\ldots ,Dr_{pd},\ldots ,Dr_{pt}\right\},\forall d\in (1,t),\forall Dr_{pd}\in \left\{0,1\right\} \end{array}$$
where $Dr_{g}$ and $Dr_{p}$ represent the vector of the drug-target gene and drug pathway, respectively, and r and t represent the length of the two vectors, respectively.

The ACS measure was an improved form of vector-based similarity. ACS can handle the relationship that has not been found yet. The influence of the relationship that has not been found on the results can be minimized by operating on the mean value. And overall, the ACS scales well. The ACS formula is as follows:(3)$$R=\frac{\sum_{i=1}^{N}(A_{i}-\bar{\alpha })(B_{i}-\bar{\beta })}{\sqrt{\sum_{i=1}^{N}{\left(A_{i}-\bar{\alpha }\right)}^{2}}\sqrt{\sum_{i=1}^{N}{\left(B_{i}-\bar{\beta }\right)}^{2}}}$$
where R is the ACS score between two diseases/drugs, $A_{i}$ is the number of corresponding positions of the gene/pathway vector of one disease/drug, $\bar{\alpha }$ is the mean value of all genes/pathways vectors of the gene/pathway list for one drug/disease, $B_{i}$ is the number of corresponding positions of the gene/pathway vector of the other disease/drug, $\bar{\beta }$ is the mean value of all genes/pathways vectors of the gene/pathway list for the other disease/drug, and N is the total length of the gene/pathway vector of the disease/drug.

2.Construction of Hybrid Matrix

On the basis of the above-mentioned homogeneity information, in this paper, the drug-similarity matrix and the disease-similarity matrix were innovatively spliced into a heterogeneous hybrid matrix in a specific way. The rows of the matrix represent disease–drug relationship pairs, which were known drugs with their indications. Columns represent drug and disease, respectively. In the column representing the drug, each position in the matrix indicates the similarity of the drug in the disease–drug relationship pair represented by that row to the drug represented by that column. In the column representing the disease, each position in the matrix indicates the similarity of the disease in the disease–drug relationship pair to the disease represented by this column. Finally, a multi-level disease–drug correlation hybrid matrix was formed with the number of disease–drug relationship pairs as the number of rows and the sum of the number of drugs and diseases as the number of columns.

#### 4.2.2. Acquisition of Drug–Disease Correlation Score

The use of support-vector regression (SVR) is the same principle as SVM for classification, with a few small differences. Supervised machine-learning models with associated learning algorithms that can analyze data for classification and regression analysis are called support-vector regression. Here, x is the drug–drug correlation score and disease–disease correlation score in the correlation hybrid matrix mentioned in the matrix above. y(y∈R) is the disease–drug correlation scores of a drug for each disease, and l is the sample size. Then the linear support-vector regression algorithm can be expressed in the following form:(4)$$y=\langle \omega ,x\rangle +\varphi ,\omega ,x\in R^{e},\varphi \in R$$

Based on the known disease–drug correlation relationships we obtained, the ω and φ of the parties in the regression equation were acquired by the learning fit of the SVR algorithm to the scores of each row in the hybrid matrix. From this, we know that, in the prediction, for the unknown pairs of relationships, the following formula is available:(5)$$f(x)=\langle \omega ,x\rangle +\varphi ,\omega ,x\in R^{e},\varphi \in R$$
where f(x) is the objective function that needs to be from the train set, 〈⋅,⋅〉 is the dot product operation in the $R^{e}$ space, and ω is the weight. The problem of solving f(x) through training can be transformed into a solution; then, the linear SVR formula can be transformed into:(6)$$\left\{ \begin{array}{l} \min\frac{1}{2}{\left\|\omega \right\|}^{2}\\ s.t.\left|\langle \omega ,x\rangle +\varphi -y_{i}\right|\leq \varepsilon ,i=1,2,\ldots ,l \end{array}\right.$$

That is, the difference between the predicted value $\left(\langle \omega ,x\rangle +\varphi_{i}\right)$ and the actual value $y_{i}$ is smaller than the constant ε (effect of constant change).

In the actual prediction process, it is often difficult to directly determine the appropriate ε to ensure that most of the data can be within the interval, and SVR expects all training data to be within the interval. So, the slack variable ξ is added, so that the interval requirement of the function changes, that is, allowing some samples to be outside the interval. After introducing the slack variables, all sample data meet the conditions:(7)$$\left|y_{i}-\left(\omega x_{i}+\varphi \right)\leq \varepsilon +\xi \right|,\forall i$$

This is the restriction after reflecting into a slack variable, so it is also called soft-interval SVR. For any sample $x_{i}$, if it is inside or on the edge of the isolation zone, ξ is 0. Above the isolation zone, ξ>0, $\xi^{*}=0$, and below the isolation zone, then $\xi^{*}>0$, ξ=0. After introducing the slack variable, the original formula can be written as:(8)$$\begin{array}{l} \underset{\omega ,b,\xi_{i},{\hat{\xi }}_{i}}{\min}\frac{1}{2}{\left\|\omega \right\|}^{2}+C\sum_{i=1}^{\delta }\left(\xi_{i},{\hat{\xi }}_{i}\right)\\ s.t.\\ f(x_{i})-y_{i}\leq \varepsilon +\xi_{i},\\ y_{i}-f(x_{i})\leq \varepsilon +\xi_{i},\\ \xi_{i}\geq 0,{\hat{\xi }}_{i}\geq 0,i=1,2,\ldots ,\delta . \end{array}$$

Here ε = 0.1, as the loss function, is an empirical value that is considered to define the minimum distance to the sample point furthest from the hyperplane.

The hybrid matrix is input into the support-vector regression machine, while the corresponding labels were given, 1 for known disease–drug relationship pairs and 0 for non-known disease–drug relationship pairs. After inputting the matrix, the SVR learns and fits a regression line, so that pairs of drug and disease relationships that satisfy the constraints of the SVR machine get predicted results and their relative degree of correlation.

In summary, after setting the parameters, the corresponding regression equations were finally obtained by fitting each row of the constructed hybrid matrix. For these diseases, the corresponding prediction models can be obtained by training on known data and adjusting the parameters. The prediction model scores the relationship between the disease and some of the drugs to be predicted separately. This score is the so-called disease–drug correlation score, which is also known as $f(x_{i})$ in the equation.

### 4.3. Screening of Potential Therapeutic Drugs

For a disease, a list of drugs was predicted by a support-vector regression algorithm after inputting the hybrid matrix into the SVR. Each drug has a correlation score with the corresponding disease. The drugs were filtered in a stepwise manner to obtain potential therapeutic drugs for the disease. The correlation scores between the drugs and a disease were calculated through random perturbation. Drug candidates for the disease were screened based on the rank of disease–drug correlation scores before and after randomized perturbation experiments, respectively. The stability score was used to measure the stability of the drug with respect to its correlation with the disease. Potential therapeutic drugs were obtained by setting up potential therapeutic criteria to screen for stable drug candidates (Appendix A).

#### 4.3.1. Drug Candidates

For a disease, the disease–drug correlation scores were calculated by SVR, and the score distribution ranged from negative to positive, with a greater positive direction indicating a stronger degree of correlation with the drug for that disease. The distribution of disease–drug correlation scores was similar for each disease.

For different diseases, the distributions of disease–drug correlation scores for the drugs were different. So, in the first step, we selected the drugs that ranked in the top 50 (de-redundant: the highest correlation score for a particular drug corresponding to a disease was taken) correlation scores as drug candidates (DrC) for a disease. Drugs ranked after the 50th were subsequently discarded.

#### 4.3.2. Stable Drug Candidates

Considering the differences in the composition of the drugs in the training set could affect the screening of drug candidates. For a given disease, correlation scores and the obtained top 50 candidate drugs changed accordingly. Therefore, a random perturbation experiment was performed. Setting stability criteria to screen drugs. Drug candidates with high stability were identified as stable drug candidates.

Three types of random perturbation processes were carried out. Training sets for multiple diseases were constructed by randomly removing data for 1, 2, and 5 different diseases. We considered the top 50 drugs in the training set of these randomization operations as drug candidates for different randomization conditions.

The frequency of the top 50 drugs was counted, and finally, the stability scores of these drugs that emerged during randomization were used to assess the stability of the drugs and obtain stable drug candidates.

For each disease, the given stability quantitative index was the stability score for a candidate drug:(9)$$S_{j}=1-\sqrt{\frac{\sum_{i=1}^{M}{\left(s_{i}-\bar{s}\right)}^{2}}{M-1}}$$
where $S_{j}$ is the stability score for a candidate drug, j indicates the type of randomization (j = 1 or 2 or 5), M is the number of times of randomization for a certain random type. s denotes the frequency of occurrence of the drug in the top 50 drugs of the list of disease–drug correlations obtained in a particular randomization of a certain randomization type. $\bar{s}$ denotes the average of the frequency of occurrence of the top 50 drugs in the list of disease–drug correlations obtained for the drug in all randomization times obtained for a certain randomization type. The larger the $S_{j}$, the more stable the frequency of the drug in the predicted results in this random perturbation.

The stability criteria were specified, and the intersection of drugs that pass the stability interval under several random perturbations with the drug candidates before they undergo perturbations is called the set of stable drug candidates $Dr_{S_{j}}$, whose set is denoted by SDrC.
(10)$$\begin{array}{c} Dr_{S_{j}}=\left\{Top95\%S_{j}\right\}\\ SDrC=Dr_{S_{1}}\cap Dr_{S_{2}}\cap Dr_{S_{5}}\cap DrC \end{array}$$

#### 4.3.3. Potential Therapeutic Drugs Identification

Considering that the stability scores of stable drug candidates fluctuate to some extent, further, the potential therapeutic criteria were set to screen the final potential therapeutic drugs.

For each disease, the stability scores are in different types of random processes. It can be derived from this:(11)$$S_{SDrC_{k}}=1-\sqrt{\frac{\sum_{i=1}^{M^{\prime }}{\left(s_{i}-\bar{s}\right)}^{2}}{M^{\prime }-1}}$$
where $S_{SDrC_{k}}$ is the total stability score of a drug in the set of stable drug candidates, and $M^{\prime }=3M+1$ and $M^{\prime }$ denotes the sum of all random counts under different random types and the number of predictions before perturbation. k denotes a drug in the set of stable drug candidates. The larger the $S_{SDrC_{k}}$, the more stable the drug appears in various conditions, and the more likely it is to be a potential therapeutic drug for the disease.

Similarly, after calculating the total stability score of a drug, the potential therapeutic interval is defined. Stable drug candidates whose total stability score of $SDrC_{k}$ in the top 95 percent were considered potential therapeutic drugs.

A collection of drugs that meet the potential therapeutic criteria is called the potential therapeutic drug set (PtDr).
(12)$$PtDr=\left\{Top95\%S_{SDrC_{k}}\right\}$$

## 5. Conclusions

In this study, we presented a novel computational strategy for drug repositioning to identify potential therapeutic drugs for a disease. The approach was successfully applied to the prediction of potential therapeutic drugs for diseases in the MESH branch. And some potential therapeutic drugs have been well-validated from different levels. Compared with traditional machine-learning methods, the DDCM strategy was able to demonstrate drug–disease correlations with continuous scores, predicting potential therapeutic drugs for the disease on a larger scale. However, by collecting large amounts of data to conduct experiments, we found that the operational efficiency of the DDCM may be affected. In the future, we will consider more aspects of drug or disease similarity to improve the confidence of our algorithm. Moreover, we can also use the drug as the subject to predict which different diseases the drug can treat, thus expanding the scope of application of our strategy. In short, DDCM provides a new perspective for the study of drug repositioning, and it can be regarded as a useful guide for drug repositioning and drug discovery.

## Figures and Tables

**Figure 1 ijms-25-05267-f001:**
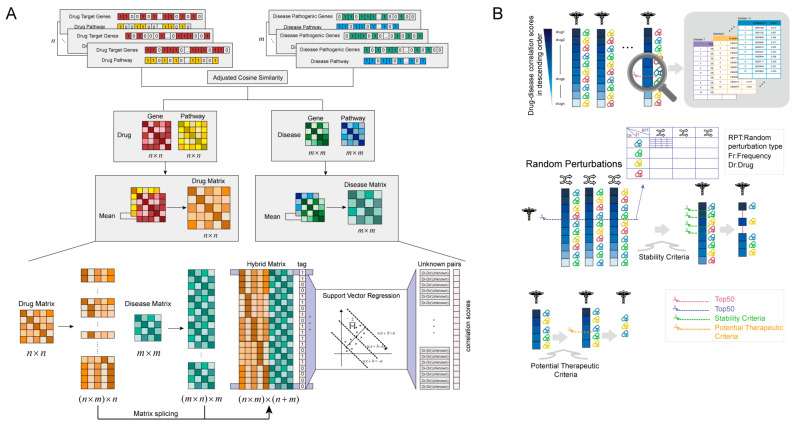
Workflow of the DDCM. (**A**) The process of forming a hybrid matrix and predicting disease–drug correlation scores in the DDCM approach. (**B**) The process of screening potential therapeutic drugs for a disease by means of a disease–drug correlation score in the DDCM approach.

**Figure 2 ijms-25-05267-f002:**
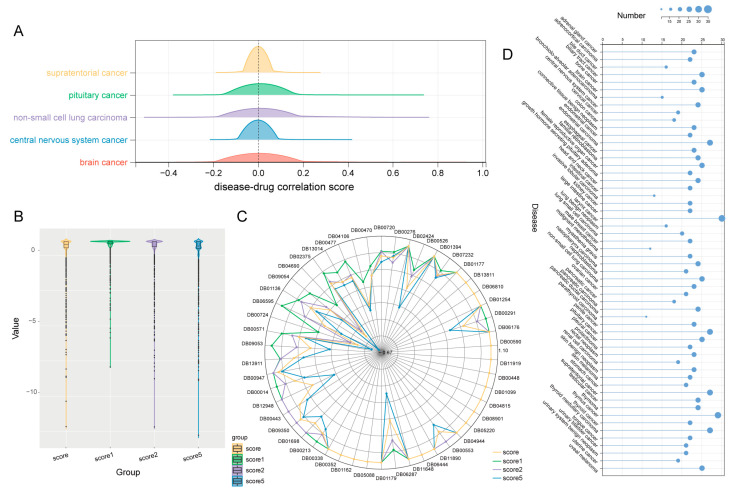
Potential therapeutic drugs for 59 neoplasms. (**A**) The distribution of disease–drug correlation scores for different diseases (The figure shows the distribution of disease–drug correlation scores for five diseases, supratentorial cancer, pituitary cancer, non-small-cell lung carcinoma, central nervous system cancer, and brain cancer). (**B**) The distribution of disease–drug correlation scores for the overall diseases in the C04.588 branch of MeSH before and after random perturbation. (**C**) Radar plot of stable drug candidates for NSCLC and the potential therapeutic drug screening criteria, where drugs with stability scores close to 1 multiple times were considered relatively stable. After the potential therapeutic screening criteria, a total of 25 potential therapeutic drugs for the final NSCLC were identified. (**D**) The number of potential therapeutic drugs for all diseases in the C04.588 branch.

**Figure 3 ijms-25-05267-f003:**
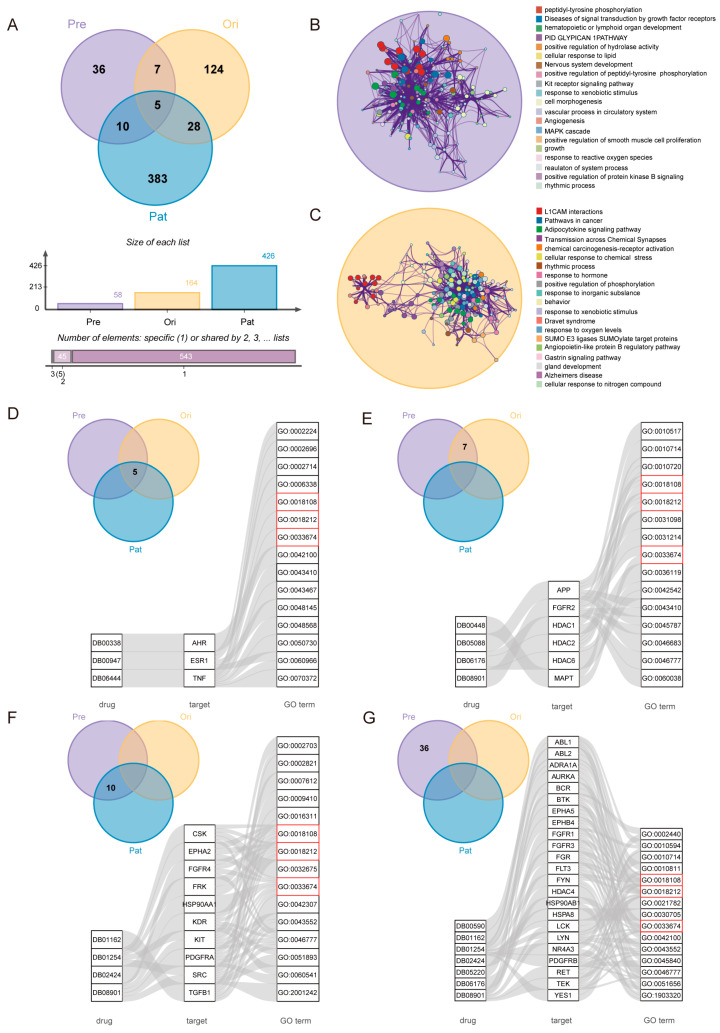
The functional aspect. (**A**) Venn diagram of drug targets of potential therapeutic drugs, drug targets of known therapeutic drugs, and pathogenic genes for NSCLC (Pre: potential therapeutic drug targets, Ori: known drug targets, and Pat: pathogenic genes of NSCLC). (**B**) Drug-target enrichment analysis of potential therapeutic drugs for NSCLC. (**C**) Drug-target enrichment analysis of known therapeutic drugs for NSCLC. (**D**–**G**) Partial Gene Ontology (GO) enrichment of drug-target subgroups for potential therapeutic drugs in NSCLC. (Some genes were not enriched on GO terms) (**D**) Group I. (**E**) Group II. (**F**) Group III. (**G**) Group IV. (GO terms enriched for all four groups of drug target genes were marked with red boxes.)

**Figure 4 ijms-25-05267-f004:**
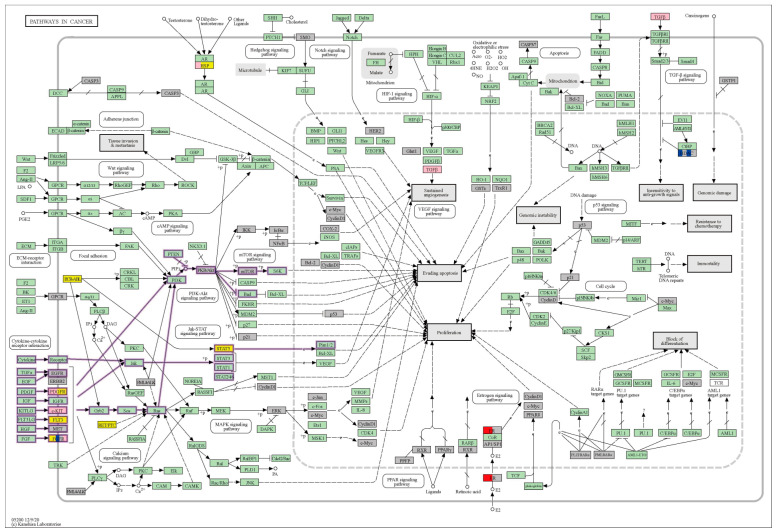
The drug-target aspect. Involvement of various groups of drug-target genes in cancer pathways as shown by KEGG Mapper. (Red: Group I; blue: Group II; pink: Group III; yellow: Group IV; grey: target genes of known drugs; pathways labeled with purple outer box lines are those in which the drug-target genes of potential therapeutic drugs are predominantly involved).

**Figure 5 ijms-25-05267-f005:**
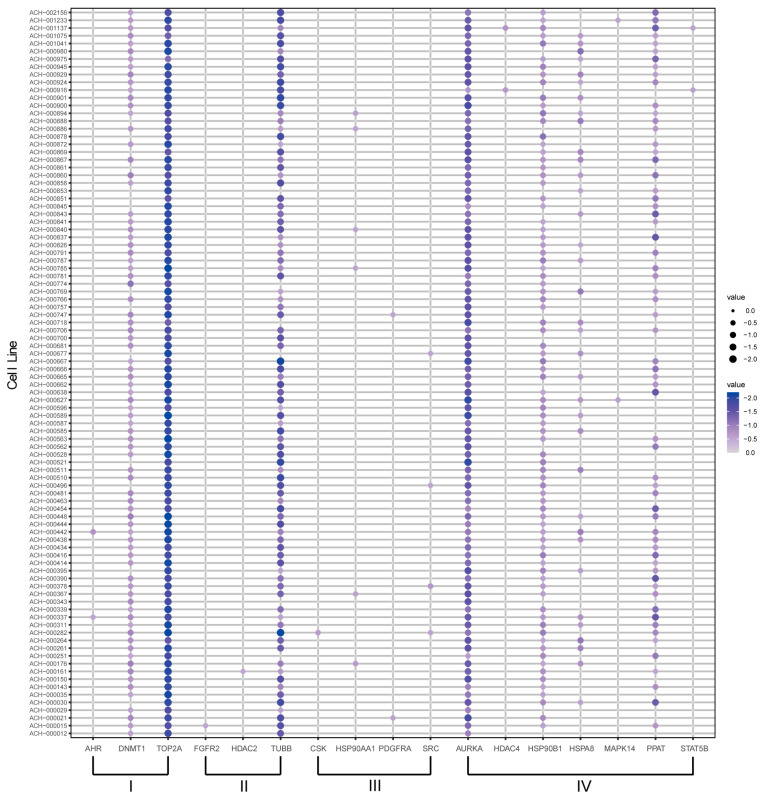
The survival-essential gene aspect. NSCLC cell lines corresponding to survival-essential genes.

**Figure 6 ijms-25-05267-f006:**
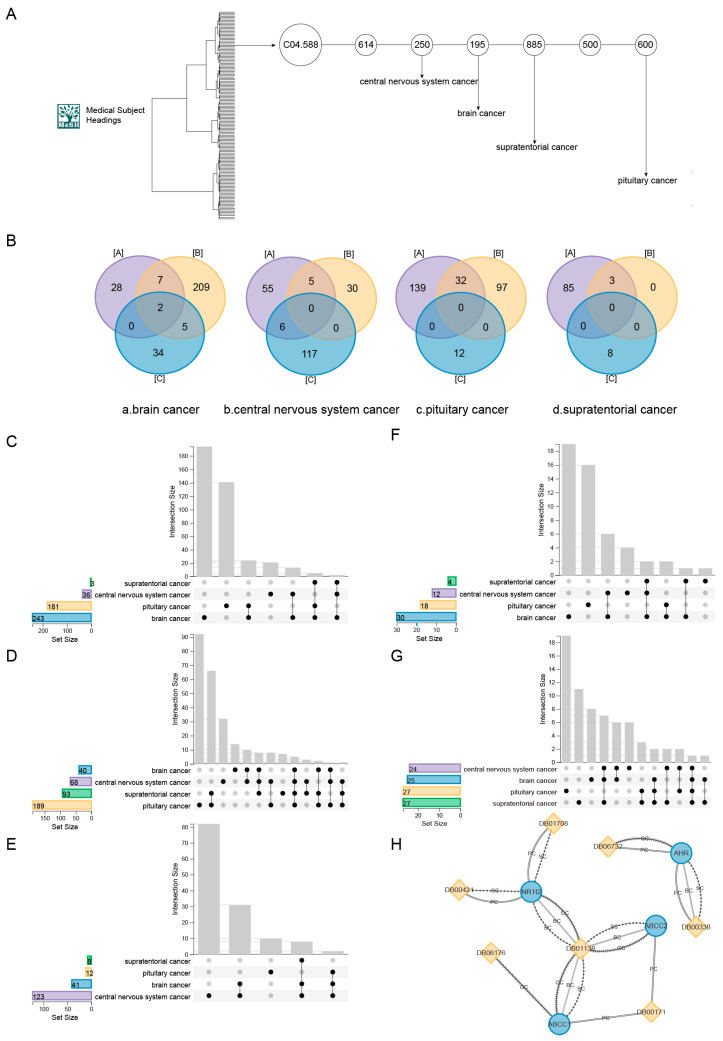
Drug research of similar neoplasms in MeSH branches. (**A**) The relationship between four cancers (brain cancer, central nervous system cancer, pituitary cancer, and supratentorial cancer) under the C04.588 branch of MeSH. (**B**) (a–d) The Venn diagrams indicate the degree of overlap of the screened potential therapeutic drug targets, drug targets of known drugs, and disease pathogenic genes for brain cancer, central nervous system cancer, pituitary cancer, and supratentorial cancer, respectively ([A] Drug-target genes of potential therapeutic drugs predicted by the DDCM approach. [B] Drug-target genes of known drugs for a disease. [C] Pathogenic genes for a disease.). (**C**–**E**) The overlap of drug targets of known therapeutic drugs, drug targets of potential therapeutic drugs, and pathogenic genes for each of the four diseases, respectively. (**F**) Overlap of known therapeutic drugs for the four diseases. (**G**) Overlap of the number of potential therapeutic drugs predicted by the DDCM method for the four diseases. (**H**) Four genes traced from four disease perspectives to predict potential therapeutic drug profiles. The blue circle indicates the gene and the yellow diamond indicates the drug. BC, CC, PC, and SC denote the four diseases (brain cancer, central nervous system cancer, pituitary cancer, and supratentorial cancer) of the retrospective approach, respectively.

**Table 1 ijms-25-05267-t001:** The drugs corresponding to the survival-essential genes in each group.

Group	Survival-Essential Genes	Drugs
I	*TOP2A, DNMT1, AHR*	DB01177/DB01179/DB01099/DB00338
II	*TUBB, FGFR2, HDAC2*	DB01179/DB01394/DB08901/DB06176
III	*SRC, PDGFRA, HSP90AA1, CSK*	DB01254/DB08901/DB02424
IV	*PPAT, STAT5B, AURKA, MAPK14, HSP90B1, HDAC4, HSPA8*	DB01254/DB05220/DB02424/DB06176

## Data Availability

The data that support the findings of this study are available upon request.

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
