# Peer review of "DDCM: A Computational Strategy for Drug Repositioning Based on Support-Vector Regression Algorithm"

_ijms, 2024, doi:10.3390/ijms25105267_

Round 1
Reviewer 1 Report
Comments and Suggestions for Authors
This manuscript proposed a novel approach integrating data from multiple sources and different levels to predict potential treatments for diseases with support vector regression. However, there are several critical issues which need to be addressed.
1. The authors proposed a new drug repurposing method to find the new therapeutic target of existing drugs. However, there is lack of an experiment for the accuracy of their algorithm. Based on the hybrid matrix of drug-disease correlative relationships, they calculated the disease-drug correlation score. Although they showed the gene sets or pathways associated with the predicted drug-disease pairs, it is very necessary to show the accuracy of the predicted drug-disease pairs. For example, the authors need to report the accuracy and ROC curve of their predictions for all pairs and per disease.
2. As the previous research pointed out, the direct overlap of target genes of a drug and causative gene of a disease is very little thus, most drug-disease pairs do not have direct overlapped genes. However current method is focused on these overlapped genes by using the binary vector with cosine similarity. Do authors check the distribution of cosine similarity of these drug, disease vectors? It seems that those vectors are very sparse and most of their cosine similarity of drug-disease pairs would be around 0. Could you please provide a cosine similarity of disease-drug pairs? If it is, how we can use this binary vector as a good metric to represent the association of disease-drug? What about the drug-disease pairs having no overlapped genes?
3. For the equations, (4) and (6) equations look same. It needs more explanation why these same equations are marked.
4. It would be good to report the predicted correlation score between drug-disease as a supplement.
5. For the stable drug candidates, random perturbation results should be attached as a supplement. For example, could you please report the distribution of stability scores?
6. Some chapters are missing. 2.1 and 2.2.1 are not in the manuscript.
Comments on the Quality of English LanguageMinor editing of English language required
Reviewer 2 Report
Comments and Suggestions for Authors
The authors of the article "DDCM: A computational strategy for drug repositioning based on support vector regression algorithm" present a well-structured and organised work based on drug repositioning using SVR algorithm strategies. In my opinion, this paper presents interesting and novel results for publication in this journal. However, I have a few minor weaknesses.
Introduction: Further clarification is required on how this study's approach differs from or improves upon existing methods.
Figure 1 should be redistributed, images A and B in a separate figure, images C, D, E and F in another figure to have more clarity on the representations and scheme of work of the article.
The conclusions of this work in a separate section highlight the results and relevance of the study.
Reviewer 3 Report
Comments and Suggestions for Authors
This manuscript presents the DCCM machine-learning methodology for drug repositioning. The authors describe their SVR methodology for uncovering new anticancer indications for approved drugs and validating the predictions against the literature; the authors also performed functional, target, and survival-essential genes analysis to validate their results on NSCLC.
However, the authors must address some concerns for clarity, such as:
- The authors must describe how they established that a gene is pathogenic for the disease.
- In the Construction of Hybrid Matrix, what does an acquired drug mean?
- The authors must consistently use the terms they introduced in the main manuscript and the supplementary materials; e.g., the "correlation score" is not used in the supplementary materials. Figure S8 (indicated in line 207) does not illustrate the correlation score of a drug with the corresponding disease.
Round 2
Reviewer 3 Report
Comments and Suggestions for Authors
The authors responded adequately to all comments and added the necessary amendments to the manuscript.